# On the Optimal Lawful Intercept Access Points Placement Problem in Hybrid Software-Defined Networks

**DOI:** 10.3390/s21020428

**Published:** 2021-01-09

**Authors:** Xiaosa Xu, Wen-Kang Jia, Yi Wu, Xufang Wang

**Affiliations:** Fujian Provincial Engineering Technology Research Center of Photoelectric Sensing Application, Key Laboratory of OptoElectronic Science and Technology for Medicine of Ministry of Education, College of Photonic and Electronic Engineering, Fujian Normal University, Fuzhou 350007, China; xiaosaxu521@163.com (X.X.); wuyi@fjnu.edu.cn (Y.W.); fzwxf@fjnu.edu.cn (X.W.)

**Keywords:** lawful interception, hybrid SDN, intercept access point, minimum vertex cover

## Abstract

For the law enforcement agencies, lawful interception is still one of the main means to intercept a suspect or address most illegal actions. Due to its centralized management, however, it is easy to implement in traditional networks, but the cost is high. In view of this restriction, this paper aims to exploit software-defined network (SDN) technology to contribute to the next generation of intelligent lawful interception technology, i.e., to optimize the deployment of intercept access points (IAPs) in hybrid software-defined networks where both SDN nodes and non-SDN nodes exist simultaneously. In order to deploy IAPs, this paper puts forward an improved equal-cost multi-path shortest path algorithm and accordingly proposes three SDN interception models: T interception model, ECMP-T interception model and Fermat-point interception model. Considering the location relevance of all intercepted targets and the operation and maintenance cost of operators from the global perspective, by the way, we further propose a restrictive minimum vertex cover algorithm (RMVCA) in hybrid SDN. Implementing different SDN interception algorithms based RMVCA in real-world topologies, we can reasonably deploy the best intercept access point and intercept the whole hybrid SDN with the least SDN nodes, as well as significantly optimize the deployment efficiency of IAPs and improve the intercept link coverage in hybrid SDN, contributing to the implementation of lawful interception.

## 1. Introduction

National security and social stability, in today’s world, have been shaken by some security threats such as terrorist attacks, cybercrime and information warfare. For the law enforcement agencies (LEAs; L), therefore, lawful interception (LI) is still one of the main means to intercept a suspect or address these illegal actions at present. As we all know, lawful interception is a kind of data acquisition of communication network based on lawful authorization for the purpose of analysis or evidence collection. Thus, it allows the law enforcement agencies with court orders or other legitimate authorities to selectively eavesdrop on individual users. Most countries require those licensed telecom operators to provide legitimate interception gateways and nodes on their networks for communication interception. To deploy the gateways and nodes in legacy networking where traditional gateways or nodes rely on dedicated devices and backhaul links to intercept network traffic, however, leads to unimaginable cost. On the contrary, software-defined networking (SDN) [1], different from the traditional networking, can simplify the traditional network’ architecture [2] and thus enable efficient management and centralized control [3] for intercepting network traffic at an extremely low cost because of its property of software definition with OpenFlow protocol [4]. The deployment of SDNs, however, is not a one-step process, but a long process, namely, in the wake of the increasing deployment of SDNs [5], a situation where both SDN nodes and non-SDN (N-SDN) nodes exist simultaneously is formed gradually. Therefore, it is of great significance to study how to design a brand-new network information lawful interception system architecture based on the software-defined network (SDN) technology and to discuss its challenges such as the deployment of intercept access point (IAPs), route selection of intercept, the minimum cost of intercept, the minimum number of intercept access points etc. in a hybrid SDN. 

In this paper, we propose the deployment and optimization strategy of intercept access points, which includes single intercept access point selection, the shortest route optimization algorithm between three points, the minimum intercept traffic cost algorithm, and the restrictive minimum vertex cover algorithm. 

The problem of single intercept access point selection is the shortest path problem that is to solve the shortest path between two given vertices in a weighted graph. At this time, the shortest path not only refers to the shortest path in the sense of pure distance, but also in the sense of economic distance, time and network. In this paper, the cost of shortest path between two points can refer to hop-count, traffic, transmission delay, transmission bandwidth, energy consumption etc. As is known to all, Dijkstra Algorithm [6] is the most typical single source shortest path algorithm, which is used to calculate the shortest path from one node to all nodes, and yet not all equal-cost multi-path shortest path. Meanwhile, Li [7] proposed an improved Dijkstra Algorithm that can find most of the shortest paths using the initial shortest path set through applying for concept of precursor node but cannot find all shortest paths. Moreover, a lot of related work with respect to the shortest path have been done by [8,9,10,11,12,13,14] in various fields.

In view of this, we develop an improved equal-cost multi-path shortest path algorithm (i.e., ECMP-Dijkstra) which can find all shortest paths between the source (S) and the destination (D), and accordingly put forward three SDN interception models based on ECMP-Dijkstra Algorithm in hybrid SDN. The three SDN interception models can be viewed as a cost-effective three-point shortest path algorithm with low time and space complexity, and thus can be used to deploy the best intercept access point reasonably in hybrid SDN.

The optimization of traffic engineering in hybrid SDN, like [15,16,17], is also one of our focuses. This study mainly concerns with the best transmission quality of intercepted data, the minimum cost of returning intercepted data to the interception center (i.e., LEA; L), the total traffic in global network, the transmission quality of traffic normally accepted by users when deploying intercept access points.

In reality, the deployment of intercept access points in the Internet does not simply corresponds to the micro perspective of a single data flow between three points. There is a very dynamic and complex traffic matrix [18] relationship and interactive influence among hundreds of millions of nodes in the large-scale Internet. A certain intercept access point (IAP; I) can meet the demand of traffic between S-D (from S to D) path, but there are also tens of millions of other traffic between intercept target node pairs, which may also flow through I node at the same time. Therefore, it is very important to select the deployment location of intercept access point, which must occupy the hub position, and greatly covers all intercepted traffic and must go through the critical path. For this reason, the location relevance of all intercepted targets and the operation and maintenance cost of operators must be taken into consideration from the global perspective, and thus the deployment problem of intercept access points is viewed as the minimum vertex cover problem (MVCP) that is NP-complete [19] to find its solution.

A lot of investigations have been done on MVCP in theory and applications for the last several decades [20,21,22]. Some parameterized algorithms about MVCP have been applied in biochemistry [23,24]. Moreover, the optimal approximation algorithm for MVCP have been proposed in [25,26,27,28,29,30]. Authors in [25,26,27,28,29,30] proposed the approximate optimization algorithm for MVCP by using the concept of degree.

Referring to their proposed algorithm, we develop a restrictive minimum vertex cover algorithm (RMVCA) in hybrid SDN networks to optimize the deployment efficiency of IAPs and to improve the link coverage of the whole interception system.

The ultimate aim of this paper is to contribute to the theory of lawful interception technology, the development of Internet and national security. In summary, the main contributions of this paper are as follows:To solve the problem of single intercept access point selection and routing between three points, we develop an improved equal-cost multi-path shortest path algorithm (i.e., ECMP-Dijkstra) and accordingly put forward three SDN interception models (e.g., T model, ECMP-T model and Fermat-point model) to deploy the best intercept access point reasonably in hybrid SDN, realizing the effective deployment of intercept access point in lawful interception system.Considering the location relevance of all intercepted targets and the operation and maintenance cost of operators of the whole interception system, we proposed a restrictive minimum vertex cover algorithm (RMVCA) to intercept the whole interception system with the least SDN nodes, optimize the deployment efficiency and improve the intercept link coverage for the whole interception system when deploying IAPs.Based RMVCA, we put forward three approaches PA, RA, and HA for experiments, and study and analyze the impact of different approaches on the efficiency of deploying intercept access points and on the intercept link coverage in hybrid SDN, to seek out the best RMVCA approach.We study and analyze the impact of different SDN interception models on various performance metrics of lawful interception system by using three real-world topologies, to seek out the best interception model.

In this paper, we first analyze various SDN interception models in hybrid software-defined networks and propose their algorithms, and then develop a restricted minimum vertex coverage algorithm from a global perspective. Extensive simulation results based on real-world network topology show that RMVCA can significantly improve network interception link coverage and deployment efficiency of IAPs of whole interception system, and that the performance metrics of the interception system are the best when Fermat-point interception model is adopted.

The remainder of this article is structured as follows. Section 2 surveys relevant work and Section 3 presents ECMP-Dijkstra Algorithm and SDN interception models. We propose the RMVCA in Section 4, followed by the performance evaluation of RMVCA and SDN interception models in Section 5. Then, Section 6 concludes the paper.

## 2. Related Works

Table 1 presents comparisons between our proposed and the related works according to different parameters.

### 2.1. Lawful Interception (LI) and Hybrid Software-Defined Networks (H-SDNs)

With the dramatic development of the Internet, an increasing number of people commit crimes on the Internet, and criminal activities are extremely rampant, which seriously affect people’s security and national stability. Thus, lawful interception (LI) is still one of the momentous means for the law enforcement agencies (LEAs) to maintain national security, crack down on crime and prevent cybercrime. For interception system, an intercept device is installed to intercept network traffic, and copies it back to LEAs, and then carries out identifying and analyzing by manual or machine. With the development of new network technology and the continuous increase of network traffic, it is a more and more common and difficult task to carry out lawful interception on the Internet [31] for helping tracking culprits and to understand the nature and behavior of current Internet traffic.

With the development of SDN technology, legacy Ether-net switches are gradually migrating to SDN, and this process is harmless [32]. Although the emerging SDN networks that provides programmability to networks can have an improvement in implementing traffic engineering (TE), management departments still hesitate to deploy SDN fully because of various reasons such as budget constraints, risk considerations as well as service level agreement (SLA) guarantees. This results in developing SDN network incrementally, i.e., to deploy the SDN network only through migrating fewer SDN switches in legacy network, thus, to form the hybrid SDN networks (H-SDNs). H-SDN network provides a coexistence and cooperation environment for N-SDN nodes and SDN nodes, which brings many benefits to traditional IP networks. For the near-optimal performance of traffic engineering, therefore, it is crucial to maximize the benefits of SDN with minimal SDN deployment. Therefore, it is imperative to deploy SDN intercept access point in a hybrid SDN (H-SDN) network where SDN nodes (routers) and legacy nodes coexist and operate in perfect harmony, realizing lawful interception. In H-SDN, the links between SDN nodes and between SDN nodes and N-SDN nodes can be intercepted (i.e., SDN links), and the links between N-SDN nodes cannot be intercepted (i.e., N-SDN links) due to the lack of special equipment and dedicated return link in hybrid SDN. In other words, in the interception system based SDN, the law enforcement agencies (LEAs) do not have to set up special equipment and a dedicated line in traditional IP networks, but can intercept traffic of links through SDN intercept access point to respond to requests from the interception center, which can greatly reduce the cost of traditional special equipment and leased lines. The interception system based SDN will be no longer restricted by the bandwidth of the intercepting dedicated equipment and link. By deploying intercept access point, the interception system will have a lot of redundant links or paths to be employed to return data flow, thus, to reduce or avoid the risk of single point failure or to further guarantee the multi-path routing method. 

Therefore, the deployment of SDN intercept access point in interception system is helpful to perfect the route of intercepting traffic, to make full use of Internet bandwidth resources, to improve user’s quality of service, and to further optimize the performance of the whole interception system.

### 2.2. Dijkstra Algorithm

The most classic single source shortest path algorithm is Dijkstra Algorithm [6], which was proposed in 1956 and became well-known three years later. Dijkstra Algorithm can calculate the shortest path from one node to all nodes, yet not all equal-cost multi-path shortest path.

Many modified algorithms based on Dijkstra Algorithm are proposed in [7,8,9,10,11,12,13,14]. An improved algorithm of Dijkstra Algorithm was proposed by Li [7]. Under the concept of precursor node, Li exploited the initial shortest path set calculated by Dijkstra Algorithm, to calculate most but not all of the shortest paths. The authors of [8] improved Dijkstra Algorithm for solving three issues, such as the ineffective mechanism to digraph. In addition, the work [9] proposed some modifications on Dijkstra Algorithm and made the number of iterations less than the number of the nodes. Work [10] proposed an optimized algorithm based on Dijkstra Algorithm to optimize logistics route for the supply chain. On the other hand, the study [11] modified Dijkstra Algorithm and the modified algorithm is very of efficiency for public transport route planning. Work [12] used Dijkstra Algorithm towards shortest path computation in navigation systems for making sensible decision and time saving decisions. By the way, the study [13] improved Dijkstra Algorithm to find the maximum load path. Work [14] introduced an improved Dijkstra Algorithm for analyzing the property of 2D grid map and increased significantly the speed of Dijkstra Algorithm. 

Referring to their proposed algorithms, we also improve Dijkstra Algorithm and propose an improved equal-cost multi-path shortest paths algorithm (ECMP-Dijkstra), which can calculate all equal-cost shortest paths from one node to all nodes, thus developing a cost-effective shortest path optimization algorithm between three points (i.e., S, D and L) with low time and space in hybrid SDN.

### 2.3. The Minimum Vertex Cover Problem (MVCP)

The traditional algorithm to solve the minimum vertex cover algorithm (MVCP) is 2-approximation [33]. This algorithm can find the set of vertex cover which is no more than twice of the optimal vertex cover, and the time complexity of the algorithm is O (E+V). More importantly, the results obtained by this algorithm are different each time, and thus may be inaccurate and not approximate solution. However, this algorithm has its advantages: every time a vertex is selected, and all the edges connected by the vertex are deleted.

The authors in [20,21,22] made much contribution to MVCP in theory and applications. The authors in [23,24] proposed parameterized algorithms for MVCP, and applied them in biochemistry. Work [25] proposed an improved greedy algorithm for minimum vertex cover problem, and the algorithm used the concept of degree (i.e., the number of links connected by a node) to carry out an order of degree and to select the node with the largest degree to add to the minimum vertex cover set until the degree of all nodes is 0 (i.e., the vertexes in the minimum vertex cover set has covered all the edges). Thus, the result is a very excellent approximate solution. However, the process of judging the degree of the algorithm is too complicated. Authors in [26] presented a greedy heuristic algorithm for MVCP to offer better results on dense graphs. The study [27] presented a breadth first search approach, which can get the exact result of MVCP for grid graphs. Work [28] proposed a near-optimal algorithm named MAMA to optimize the unweighted MVCP, and MAMA can return near optimal result in quick-time. Authors in [29] proposed a NHGA for MVCP to yield near-optimal solutions. In [30], authors studied an ameliorated genetic algorithm for the partial VCP to skip the local optimum by powerful vertex and adaptive mutation. All of their algorithm are based on the concept of degree.

Combining with the advantages of the above algorithms, we proposed an ameliorated restrictive minimum vertex cover algorithm (RMVCA) in hybrid SDN using the concept of degree to significantly simplify the process of degree judgment and to yield near-optimal result, thus, in the whole interception system, realizing the optimization of the deployment efficiency of IAPs and the improvement of intercept link coverage.

## 3. ECMP-Dijkstra Algorithm and SDN Interception Models

### 3.1. ECMP-Dijkstra Algorithm

When deploying the best intercept access point in hybrid SDN, we have to calculate all equivalent shortest paths between two points and then select out the best route from all equal-cost shortest paths to choose the best node as IAP. The most typical single source shortest path algorithm is Dijkstra Algorithm [6]. Accordingly, an improved algorithm of Dijkstra Algorithm was proposed by Li [7]. Under the concept of precursor node, Li exploited the initial shortest path set calculated by Dijkstra Algorithm, to calculate most but not all of the shortest paths. In view of this, on the basis of Dijkstra Algorithm and Li’s Algorithm, we propose an improved equal-cost multi-path shortest paths algorithm (ECMP-Dijkstra), which can calculate all equal-cost shortest paths from one node to all nodes. The notations used in the algorithms and in the following equations are listed in Table 2.

The pseudo code of Dijkstra Algorithm is given in Algorithm 1. We input the source node *s* and an undirected graph *G (V,E)* where *V* denotes the set of all nodes and *E* denotes the set of all edges. We explain Algorithm 1 that inf denotes an infinity and *sps,i* denotes the shortest path from the source node *s* to node *i*. In lines 11–14, we get the minimum hop-count value minhops and the corresponding node key. In lines 15–19, we remove node key from U and then add node key to S and add node key to the shortest path sps,i to get the shortest path sps,key. Finally, we obtain the shortest path set SP from the source node s to all nodes in V.
**Algorithm 1** Dijkstra Algorithm**Input:***s*; *G(V,E)***Output:***SP* 1: *S(s)* = 0; *U(i) = inf, i* ∈ V, i ≠ s; *SP = ∅* 2: *SP*
*←*
*SP* ∪ *sp_s,s_* 3: **while**
*U* ≠ *∅* do 4:  *tsp =*
*∅**; minhops= inf; key = None*5:  **for** edge *e_i,j_* in *E*
**do** // node *i, j* ∈ V, i ≠ j 6:   **if**
*hops(e_i,j_)* + *S(j)* ≤ *U(i)*
**then** 7:    *U(i)*
*←**hops(e_i,j_) + S(j)* 8:    *tsp(i) = j*
 9:   **end if**10:  **end for**11:  *numu(k), inu(k)*
*←**sort(U(i))*12:  *β*
*←*
*Num(numu(k))*13:  *minhops= numu(β)*
14:  *U* ← *U* ─ *key*15:  *S(key) = minhops*16:  *S(key) = minhops*17:  **for** shortest path *sp0_s,i_* in *SP*
**do**18:   **if**
*i == tsp(key)*
**then**19:    *sp_s,key_ ← Merge(sp_s,i_,key)*
20:    *SP ← SP* ∪ *sp_s,key_*21:   **end if**22:  **end for**23: **end while**
24: **return**
*SP*

Based on Dijkstra Algorithm, we propose an improved equal-cost multi-path shortest paths algorithm (ECMP-Dijkstra) so as to calculate all equal-cost shortest paths from the source node *s* to all nodes. Detailed pseudo code of ECMP-Dijkstra Algorithm is summarized in Algorithm 2. At beginning, we input the source node *s* and the shortest path set *SP* calculated by Dijkstra Algorithm, which contains only one shortest path from *s* to all nodes. In line 1, we use the shortest path set *SP* to calculate the minimum hop-count or cost set *S* from *s* to all nodes by the function *hops()* and *S(i)* denotes the minimum cost from node *s* to node *i*. In line 5, *rsp(i)* denotes all equal-cost shortest paths from the source node *s* to the destination node *i*. We loop through the edge-set *E(ei,j)* and judge whether the hop-count or cost from node *s* to node *i* (i.e., *S(i)*) plus the hop-count of *edgei,j* equals the hop-count from node *s* to node *j* (i.e., *S(j)*). If it does, then we add node *j* to all equal-cost shortest paths from node *s* to node *i* in lines 11–12, thus obtaining multiple shortest paths from node *s* to node *j* and adding them to the shortest path set *SP* in line 13. In lines 2–13, we exploit the precursor node and the initial shortest path set *SP* repeatedly, to add equal-cost shortest paths to *SP* and thus update *SP* constantly. In line 18, we delete the duplicate shortest path from *SP* using the function *DeleteDup()*. Thus, we update the shortest path set *SP* repeatedly until the number of shortest paths in SP does not increases.
**Algorithm 2** ECMP-Dijkstra Algorithm**Input:***s; G(V,E); SP***Output:***SP* 1: *S*
*←*
*hops(SP)* 2: **repeat** 3:  *nSP*
*←*
*Num(SP)* 4:  **for** shortest path *sp_s,i_* in *SP* do 5:   rsp(i) = *sp_s,i_*
// *sp_s,i_* may contain more than one shortest path. 6:  **end for** 7:  **for** edge *e_i,j_* in *E* do // node *i, j* ∈ V, i ≠ j 8:   *sp0_i,j_*
*←*
*Sp(e_i,j_)*
// Convert *e_i,j_* to shortest path *sp0_i,j_*. 9:   **if** shortest path *sp0_i,j_* ∉ *SP*
**then**10:    **if**
*S(i)* + *hops(sp0_i,j_)* == *S(j)*
**then**11:     **for** shortest path *sp’_s,i_* in *rsp(i)* do12:      *sp’_s,i_ ←* Merge(*sp’_s,i_*, *j*) 13:      *SP*
*← SP* ∪ *sp’_s,i_*14:     **end for**15:    **end if**16:   **end if**17:  **end for**18:  *SP*
*←*
*DeleteDup(SP)*19:  *nSP’*
*←*
*Num(SP)*20: **until**
*nSP* == *nSP’*

We use three real-world topologies CRN, COST 239, NSFNet for simulation experiments, where China’s 156 major railway nodes network (China Railway Network; CRN) [34] has 156 nodes and 226 links, Pan-European fiber-optic network (COST 239) [35] has 28 nodes and 41 links and T1 NSFNet network topology [36] has 14 nodes and 21 links. Under the three topologies, we compared ECMP-Dijkstra Algorithm with Dijkstra Algorithm and Li’s Algorithm and the experimental results are shown in Figure 1 where TSP denotes the total number of shortest paths from one node to all nodes. Moreover, the higher the TSP, the better the intercept access points deployment may be. From the figures, we know that TSP of ECMP-Dijkstra Algorithm is higher than Dijkstra Algorithm and Li’s Algorithm, thus, to deploy intercept access point reasonably.

### 3.2. SDN Interception Models

For lawful interception in hybrid SDN, we first need to analyze how to intercept, that is, how to deploy intercept access point between the source (S), the destination (D) and the interception center (L). In this section, we will analyze various network interception models (i.e., the deployment strategies of IAP) in hybrid SDN. The deployment of intercept access point includes the single “IAP selection problem” in the shortest path S-D (i.e., the shortest path from S to D) and its derived “the shortest path algorithm problem between three points (i.e., S, D and L)”. The above two problems can be viewed as the same problem. Once the location of the intercept access point is determined, then the fourth point (IAP; I) can meet the service traffic between S, D and L. Under the condition that S-I, D-I, and L-I path are the shortest at the same time, the shortest path between three points can be solved to meet the needs of interception system. 

We aim to solve the problem of selecting single intercept access point and routing between three-points, namely to deploy the best intercept access point in the shortest paths between S, D, and L. Analyzing interception models in hybrid SDN, we divide them into two interception models by the deployment location of intercept access point: legacy interception models and SDN interception models as shown in Figure 2 and Figure 3.

The legacy interception models include: S/I model, D/I model, and L/I model as shown in Figure 2a–c. As we all know, the interception service in legacy networks is limited by the deployment location of intercept access point due to the unimaginable cost of setting up special equipment and dedicated return link, to intercept network traffic. Thus, S or D or L is usually adopted as intercept access point I used to respond to the requirements of the interception center and to perform the traffic interception action in legacy network.

In this paper, we mainly study and analyze the SDN interception models, which includes T model, ECMP-T model and Fermat-point model as shown in Figure 3a–c. In view of the performance metrics of lawful interception system, the three SDN interception models are used to thoroughly study to find the optimal algorithm of deploying intercept access point. 

Figure 3a shows T model: its name comes from the topology similar to the T-word. Under the concept of SDN networking in an undirected and weighted network *G(V,E)*, any SDN node on the shortest path S-D can be selected as intercept access point (I) under the premise of not affecting the existing shortest path arrangement of S-D (i.e., maintaining the existing end-to-end transmission quality). While only the node with the minimum hop-count (or cost) to the interception center (L) should be adopted as the best I-point to run the function to capture traffic transferred to the interception center. 

Figure 3b shows ECMP-T model: based on the operation mode of T model, the path I-L must be the shortest path, but this shortest path S-I-D does not necessarily meet the optimal path. In fact, there may be more than one shortest path S-D, namely, the shortest path S-D is equal-cost multi-path (ECMP). Hence there may be a southward equal-cost shortest path in the T-word path theoretically, in which there is another intercept access point (I) and the hop-count (or cost) of I-L path is lower than the current one, so this interception model is called ECMP-T model that the nearest I-point from the interception center (L) is selected as the best intercept access point I among all the equivalent shortest paths between S and D.

Detailed pseudo code of T or ECMP-T model is presented in Algorithm 3. At the beginning of the algorithm, the set I used to store the best intercept access point is set to be empty in line 1. In lines 2–4, we calculate the shortest path *sp_S,D_* from *S* to *D* using Dijkstra or ECMP-Dijkstra Algorithm and then obtain the node-set N_S,D_ in the shortest path *sp_S,D_*, and next select out SDN nodes from the node-set *N_S,D_* to get the SDN node-set *SN*. If the SDN node-set *SN* is not empty, we traverse SDN nodes in *SN* and implement lines 6–16; otherwise, we fail to deploy intercept access point (IAP) between *S*, *D* and *L*, and thus save the wrong node combination of *S*, *D* and *L* in line 18. Line 7 calculates the lowest hop count (or cost) from SDN nodes to L and get the cost vector *h(i).* In line 9, we sort the cost vector *h(i)* by size of hop count in descending order and then get the sorted vector *numh(j)* and the corresponding label vector *inh(j)*, where j denotes the subscript of j-th element of a vector. Line 11 takes the minimum hop-count value *minhops* from the sorted vector *numh(j)*. Finally, in lines 13–14, we select the node with the minimum cost *minhops* as the best intercept access point and then add the selected IAP *inh(j)* to the set I.
**Algorithm 3** T or ECMP-T Model**Input:***N_SDN_*; *S*; *D*; *L***Output:***I* 1: *I* = *∅* 2: *sp_S,D_*
*←*
*Dijkstra(S,D)* or *ECMP-Dijkstra(S,D)* 3: *N_S,D_*
*←*
*Onodes(sp_S,D_)* 4: *SN*
*←*
*Select(N_S,D_, N_SDN_)* 5: **if** the SDN node-set *SN* ≠*∅*
**then**
 6:  **for** node *i* in *SN*
**do**
 7:   *c(i)**←**hops(Dijkstra(i,L)) or*
   *hops(ECMP-Dijkstra(i,L))*
 8:  **end for**
 9:  *numh(j), inh(j)*
*←*
*sort(h(i))*10:  *β*
*←*
*Num(SN)*11:  *minhops**←*
*numh(β)*12:  **for** key *j* in *numh*
**do**
13:   **if**
*numh(j)* = *minhops*
**then**
14:    *I* ← *I* ∪ *inh(j)*15:   **end if**16:  **end for**17: **else**18:  *SaveFail(S,D,L)*19: **end if**20: **return *I***

The only difference of pseudo code of T model and ECMP-T model is whether to use Dijkstra Algorithm or ECMP-Dijkstra Algorithm to calculate the shortest path. 

Figure 3c shows Fermat-point model: In geometry, Fermat-point refers to the point with the smallest sum of the distances from the three vertices of the triangle. Accordingly, we extend it to the node with the smallest sum of the distances from the three nodes of S, D and L in SDN network, and at the same time with meeting the constraints of the shortest path of S-D, S-L and D-L between the three points. Theoretically, Fermat-point model is optimal. 

Details of pseudo code of Fermat-point model are summarized in Algorithm 4. In lines 2–4, we calculate all equal-cost shortest paths of S-D, S-L, D-L using ECMP-Dijkstra Algorithm, and then obtain all node sets in the equal-cost shortest paths in lines 5–7, and next combine these node sets to get the node-set *N_S,D,L_* in line 8, and further select out SDN nodes from the node-set *N_S,D,L_* to get the SDN node-set *SN*. If the SDN node-set *SN* is not empty, we traverse SDN nodes in *SN* and implement lines 11–24; otherwise, we fail to deploy intercept access point (IAP) between S, D and L, thus to save the wrong node combination of S, D and L. Lines 12–14 calculate the lowest hop count (or cost) of i-S, i-D, i-L, and then add the results to the sum, to get the cost vector *h(i)* in line 15. In lines 17–24, we sort the cost vector *h(i)* by size of cost value in descending order and then take the minimum cost value *minhops*, and next select the node *inh(j)* with the minimum cost *minhops* as the best intercept access point and finally add the selected IAP *inh(j)* to the set *I*.
**Algorithm 4** Fermat-point Model**Input:***N_SDN_*; *S*; *D*; *L***Output:***I* 1: *I* = *∅* 2: *sp_S,D_,sp_S,L_,sp_D,L_**←**ECMP-Dijkstra((S,D),(S,L),(D,L))*3: *N_S,D_, N_S,L_, N_D,L_**←*
*Onodes((sp_S,D_, sp_S,L_, sp_D,L_)*4: *N_S,D,L_*
*←**N_S,D_* ∪*N_S,L_* ∪*N_D,L_* 5: *SN*
*←*
*Select(N_S,D,L_, N_SDN_)*6: **if** the SDN node-set *SN* ≠ *∅*
**then**
7:  **for** node *i* in *SN*
**do**
8:  *hs(i)*
*←*
*hops(ECMP-Dijkstra(i,S))*9:  *hd(i)*
*←*
*hops(ECMP-Dijkstra(i,D))*10:  *hl(i)*
*←*
*hops(ECMP-Dijkstra(i,L))*11:  *h(i)*
*←*
*hs(i) + hd(i) + hl(i)*12:  **end for**
13: *numh(j), inh(j)*
*←*
*sort(h(i))*14:  *β*
*←*
*Num(SN)*15:  *minhops*
*←*
*numh(β)*16:  **for** key *j* in *numh***do**
17:   **if**
*numh(j)* = *minhops*
**then**
18:    *I* ← *I* ∪ *inh(j)*19:   **end if**20:  **end for**21: **else**22:  *SaveFail(S,D,L)*23: **end if**24: **return**
*I*

We use *sp_i-j_* to denote the shortest path from node *i* to node j, and *hops_i-j_* denotes the lowest hop-count or cost from node *i* to node *j*. We use ‘→’ to denote that the next-node is N-SDN node and use ‘⇒’ to denote that the next-node is SDN node. Examples of three interception models are illustrated in Figure 4, where we select node 154, node 9, node 105 all marked by red as S, D and L respectively and select 30 nodes randomly in Figure 4 as SDN nodes which includes node i∈{4, 8, 11, 19, 23, 25, 31, 38, 49, 50, 58, 60, 65, 67, 77, 82, 89, 92, 100, 103, 117, 120, 121, 125, 128, 134, 140, 150, 152, 156}, to construct a hybrid SDN.

We run T interception model: One shortest path from node 154 to node 9 is *sp_154-9_* marked by pink in Figure 4 that is 154 → 153 ⇒ 152 → 146 → 142 → 136 ⇒ 134 → 124 ⇒ 121 ⇒ 117 → 94 → 81 ⇒ 82 → 74 → 52 ⇒ 49 → 32 → 30 ⇒ 31 ⇒ 25→9, and *hops_154-9_* = 20. Among all nodes in *sp_154-9_*, node 117 that is an SDN node has the lowest hop count to node 105 due to *hops_117-105_* = 6, and thus node 117 can be used as the best intercept access point I in T interception mode.

We run ECMP-T interception model: There are 22 equivalent shortest paths from node 154 to node 9, but we only show three shortest paths (i.e., *sp_154-9_* contains *sp1_154-9_*, *sp2_154-9_*, and *sp3_154-9_*) from node 154 to node 9 marked by pink, bright green, turquoise respectively in Figure 4. *sp1_154-9_* is 154 → 153 ⇒ 152 → 146 → 142 → 136 ⇒ 134 → 124⇒ 121 ⇒ 117 → 94 → 81 ⇒ 82 → 74 → 52 ⇒ 49 → 32 → 30 ⇒ 31 ⇒ 25 → 9, and *sp2_154-9_* is 154 → 153 → 155 → 144 ⇒ 140 → 133 ⇒ 134 → 124 ⇒ 121 ⇒ 117 → 99 → 97 → 69 → 68 → 61 ⇒ 60 → 56 ⇒ 23 → 24 ⇒ 25→9, and *sp3_154-9_* is 154 → 153 → 155 → 144 ⇒ 140 → 133 ⇒ 134 → 115 → 113 → 112 ⇒ 100 → 101 → 64 → 63 → 62 → 59 → 17 → 16 ⇒ 19 → 10 → 9, and *hops_154-9_* = 20. Among all nodes in *sp_154-9_*, node 100 that is SDN node in *sp3_154-9_* has the lowest hop count to node 105 due to *hops_100-105_* = 4, and thus node 100 can be used as the best intercept access point (I) in ECMP-T interception mode. Apparently, *hops_100-105_* < *hops_117-105_*, namely, this I-point outperforms the one in the T model.

We run Fermat-point interception model: the node-set *N_154-9-105_* with no repeat is obtained by all sp*_154-9_*, sp*_154-105,_*sp*_9-105_* (i.e., sp_S-D_, sp_S-L_, sp_D-L_). Namely, *N_154-9-105_* contains all nodes of all the shortest paths from node 154 to node 9, from node 154 to node 105, and from node 9 to node 105. And then, the sum of hop-count from node 103 in *N_154-9-105_* to node 154, node 9, node 105 (i.e., *hops_103-154,9,105_*) is the smallest and *hops_103-154,9,105_* = 23, This means that node 103 that is SDN node in *N_154-9-105_* can be used as the best intercept access point I in Fermat-point interception mode.

We have solved the problem of single intercept access point deployment above, and then expand to deploy intercept access points in hybrid SDN. 

Running different network interception models, we will study and analyze the influence on the best transmission quality of intercepted data (the minimum cost from intercept access point (I) to interception center (L); MILC), the total cost of running intercept operation in global network (TOC), and the quality of service of normal user’s data stream (UQoS) with different proportion of SDN node. According to the proposed three models in Figure 3, MILC, TOC, UQoS are calculated in respectively in (1), (2) and (3), where *N* denotes the maximum node label or index, and any node can be selected as S, D and L in hybrid SDN topology, i.e., there are *N^3^* possibilities for node-combination of S, D and L. After the node-combination selection (S, D, L), the best intercept access point (I) can be got by the SDN interception models, then the hop count or cost of the shortest path S-I, D-I and L-I can be calculated by the function *hops(i,j)*, thus calculating MILC, TOC and UQoS.

## 4. Restricted Minimum Vertex Cover Algorithm

There is no exception that most network optimization deployment problems can be viewed as the minimum vertex cover problem (MVCP) in graph theory. In the process of migration of SDN technology for large-scale Internet, it may be faced with the situation of hybrid deployment of SDN nodes and non-SDN nodes (N-SDN). In this hybrid SDN, not all nodes have software-defined functions to play the role of intercept access point. Only some nodes with the function of software definition can respond to the requirements of the interception center and to run interception operation. Therefore, it is very critical to select the best deployment location of intercept access point. And IPA must occupy the position of the hub, greatly covering all traffic through the critical path, and under a certain proportion of threshold, it may not achieve 100% intercept link coverage. Therefore, the minimum vertex cover problem must be transformed into the restricted minimum vertex cover problem question (RMVCP).
(1)$$\mathrm{MILC}=\sum_{S=1}^{N}\;\sum_{D=1}^{N}\;\sum_{L=1}^{N}hops(L,I)$$
(2)$$\mathrm{TOC}=\sum_{S=1}^{N}\;\sum_{D=1}^{N}\;\sum_{L=1}^{N}hops(S,I)+hops(D,I)+hops(L,I)$$
(3)$$\mathrm{UQoS}=\sum_{S=1}^{N}\;\sum_{D=1}^{N}\;\sum_{L=1}^{N}hops(S,I)+hops(I,L)$$

Considering overall situation (e.g., the location relevance of all intercepted targets, the operation and maintenance cost of operators) from the whole interception system, we intend to develop a restricted minimum vertex cover algorithm (RMVCA) to achieve the best intercept link coverage of the whole network with the minimum number of intercept access points as well as optimize the efficiency of deployment when deploying intercept access points in the hybrid SDN.

RMVCP: given a network graph *G(V,E)*, where *V* denotes the set of all nodes, and E denotes the set of all links in the network. There exists non-SDN nodes and SDN nodes at the same time in the network where V = S∪N, and *S* denotes the set of SDN nodes, and N denotes the set of non-SDN nodes. To find a *P* set (*P* ⊆ *S* ⊆ *V*), so that every link in the network is covered (intercepted) by at least an SDN node in the *P* set.

Figure 5 shows an example of solving RMVCP. In this hybrid SDN, SDN nodes (i.e., solid circle) set *S* = {1, 3, 8, 9, 11, 12, 13, 17, 19, 20, 21, 22, 25, 26, 27, 28} and non-SDN nodes (i.e., light circle) set *N*= {2, 4, 5, 6, 7, 10, 14, 15, 16, 18, 23, 24}. Using RMVCA, the SDN nodes set *P* = {1, 8, 9, 11, 13, 20, 22, 25, 26, 27} is recommended to be selected as the intercept access points set, but 7 links (marked as dotted lines) in the example failed to be covered due to the hybrid deployment of SDN and N-SDN nodes, and thus only about 80% of the links (marked as solid lines) are completely covered by 10 intercept access points. 

RMVCA ensure the result a near-optimal solution or one of the approximate solutions, so as to meet the optimal solution of the deployment problem of intercept access points. 

Based on the concept of degree, we, at a time, use greedy algorithm to select one approximate or equivalent optimal intercept access point to reduce the scale of the problem recursively, so as to obtain the minimum vertex approximation set covering all SDN links and achieve the best intercept link coverage with the minimum number of intercept access point. 

Details of pseudo code of RMVCA are summarized in Algorithm 5. We input an undirected and weighted network cover set *P* is set to empty originally. In line 2–3, we get the set *N* of N-SDN nodes and the accordingly edge-set *E_N-SDN_* of N-SDN nodes by the set *N*. Line 4 removes *E_N-SDN_* from the edge-set E, to get the edge set *E_SDN_* of SDN nodes. Lines 6–7 traverse each SDN node and calculate its degree *d(i)*. In line 9, we sort the degree vector *d(i)* in the ascending order and get the sorted degree vector *numd(j)* and the accordingly label vector *ind(j)* where j denotes the index or subscript of j-th element. Line 11 selects the maximum degree *numd(β)* from *numd(j)* where *β* denotes the number of SDN nodes in the set *S*. In lines 12–18, we judge the degree of node and implement accordingly measures. If the maximum degree of node is not equal to zero in line 12, we first add the node *ind(β)* where *β* denotes the subscript of β-th element to the minimum vertex cover set *P* in line 13 and then calculate the adjacent edge-set *θ(ind(β))* of node *ind(β)* in line 14, and next remove the adjacent edge-set *θ(ind(β))* from the edge-set *E* in line 15, which leads to the degree reduction of each SDN node. Finally, we return line 5 to judge whether *E_SDN_* is empty and then calculate the degree of each SDN node again. Otherwise, if the maximum degree of node is equal to zero, we break the loop and end the algorithm.
**Algorithm 5** RMVCA (Restricted Minimum Vertex Cover Algo-rithm)**Input:***G(V,E)*; *S*: the SDN node-set**Output:***P* 1: *P* = *∅* 2: *N* ← *V* ─ *S* 3: *E_N-SDN_*
*←*
*Edge(N(i))* 4: *E_SDN_*
*←*
*E ─E_N-SDN_*
5: **while**
*E_SDN_* ≠ *∅*
**do**6:  **for** node *i i*n *S*
**do** 7:   *d(i)* ← *i* 8:  **end for**
 9:  *numd(j), ind(j)*
*←*
*sort(d(i))*10:  *β*
*←*
*Num(S)*11:  *maxnumd*
*←*
*numd(β)*12:  **if**
*maxnumd* > *0*
**then**
13:   *P* ← *P* ∪ *ind(β)*
14:   *θ(ind(β))*
*←**ind(β)*15:   *E_SDN_*
*←**E_SDN_ ─ θ(ind(β))*16:  **else**17:   break18:  **end if**19: **end while**20: **return**
*P*

Using RMVCA proposed above, we will study and analyze the influence of different SDN node proportion on the maximum intercept link coverage of the whole network (i.e., max-ILC) and the accordingly needed minimum number of SDN nodes in the P set for realizing the maximum intercept link coverage (i.e., numP), as well as the influence of RMVCA on the intercept link coverage (i.e., ILC) and the efficiency of deploying intercept access points in whole hybrid SDN.

## 5. Simulation and Results

### 5.1. Simulation Environment and Performance Metrics of Lawful Interception System

In our simulation, we choose three real-world backbone topologies CRN, COST 239, NSFNet to evaluate the performance of three SDN interception models. Under the three network topologies, we randomly select different number of nodes as SDN nodes to construct the hybrid SDN network and the weight of each link is set to 1 by default, and the source node (S), the destination node (D) and the interception center (L) are selected randomly and thus there are 3,796,416 (156^3^), 21,952, 2,744 node combinations of S, D and L.

Under different proportion of SDN nodes, we will study and analyze the influence of different SDN interception models on the best transmission quality of intercepted data (the minimum cost from intercept access point (I) to the interception center (L); MILC), the total cost of running intercept operation in global network (TOC), and the quality of service of normal user’s data stream (UQoS), the deployment efficiency of IAP (the total number of times to calculate the shortest path during the process of deploying IAP; TTC)), and the total number of failures to deploy IEP (i.e., NFD). 

According to the proposed three SDN interception models, MILC, MRLC, TOC, and UQoS are calculated respectively in (1), (2) and (3). Based RMVCA, we run different SDN interception models and calculate and count up MILC, TOC, UQoS, TTC and NFD of each node combination of S, D and L and then compare and analyze the results to evaluate the performance of three SDN interception models.

### 5.2. Benchmark Approach

In order to analyze the influence of RMVCA on the intercept link coverage of whole hybrid SDN and the efficiency of deploying intercept access points, we propose three approaches, proactive approach (PA), reactive approach (RA), hybrid approach (HA), and then compare them by running three SDN interception models in real-world topology CRN. To show the effectiveness of HA, we compare it with the following baselines: PA and RA. 

Experimental initialization: We randomly select some nodes as SDN nodes (i.e., given a hybrid SDN network topology), and then use RMVCA to calculate the minimum vertex cover set P required to achieve the maximum intercept link coverage in theory and the accordingly number N of SDN nodes in the P set. Additionally, the calculation amount of this initialization process is negligible compared with the one of the whole H-SDN. 

Nodes selection: we traverse any node as S, D and L in topology CRN (i.e., there are 3,796,416 (156^3^) possibilities for node-combination of S, D, L) and then the node combination of S, D and L is given for experiments. 

Proactive approach (PA): when running SDN interception models to deploy intercept access point, we select the best intercept access point from the minimum vertex cover set P calculated by RMVCA. Details of pseudo code of PA in T or ECMP-T model are summarized in Appendix A. The only difference of pseudo code of PA in T model and ECMP-T model is whether to use Dijkstra Algorithm or ECMP-Dijkstra Algorithm to calculate the shortest path. 

Reactive approach (RA): according to the selected node combination of S, D and L, we run three interception models without exploiting RMVCA to deploy intercept access points.

Hybrid Approach (HA): running three SDN interception models to deploy intercept access point, we get the node-set *N_S,D,L_* where all nodes are selected from the shortest paths between S, D and L, and then obtain the node-set SP whose nodes also exist in the node-set P calculated by RMVCA. If the node-set *SP* is not empty, we preferentially select node from the *SP* set to deploy the best intercept access point; otherwise, we implement RA. Details of pseudo code of HA in T or ECMP-T model are summarized in Appendix A. 

When implementing PA or RA or HA, we count and calculate the frequency of the nodes selected as the best intercept access point, and then sort the nodes from largest to smallest based their frequency, and next select the first N nodes and calculate their intercept link coverage for studying and analyzing the impact of different approaches on the intercept link coverage (i.e., ILC) of the whole hybrid SDN. Additionally, we count the total times of calculating the shortest path (i.e., TTC) during the process of deploying intercept access points for studying and analyzing the impact of different approaches on the efficiency of deploying IAPs.

### 5.3. Results and Discussion

#### 5.3.1. ILC

Using RMVCA, we study and analyze the influence of different numbers of N-SDN nodes on the maximum intercept link coverage (i.e., max-ILC) and the accordingly needed minimum number of SDN nodes in the P set for realizing the maximum intercept link coverage (i.e., numP). Moreover, we take the operator’s operation and maintenance cost (i.e., the minimum number of SDN nodes) and network intercept link coverage into account comprehensively, so as to find the best proportion of SDN nodes from the experimental results.

Randomly selecting the number of N-SDN nodes (node i ∈ (0,156)) in CRN topology, we conducted 10,000 experiments in the same proportion of N-SDN nodes. Due to the different network topologies under the same SDN node proportion, the results of each experiment are different. The statistical results of 10,000 experiments are shown in Figure 6 and Figure 7. 

Figure 6 shows the influence of different numbers of N-SDN nodes on max-ILC. From the figure, we can see that the number of SDN links in hybrid SDN decreases gradually with the increase of the number of N-SDN nodes (the decrease of the number of SDN nodes), resulting in the gradual decline of the network intercept link coverage. And max-ILC = 0.00% denotes that all links in the whole network are N-SDN links that cannot be intercepted, namely, all nodes in the network are N-SDN nodes. Additionally, we can see that the intercept link coverage of the whole hybrid SDN can reach 80.53~100% when the number of N-SDN nodes is between 0 and 57 (i.e., the number of SDN nodes is between 99 and 156), namely, only when the number of SDN nodes in hybrid SDN is more than 99 can SDN nodes intercept more than 80% of the links of the whole network.

Figure 7 shows the influence of different numbers of N-SDN nodes on numP. From the figure, we can see that when the number of N-SDN is 0 (i.e., the number of SDN nodes is 156), 79 SDN nodes are required to achieve the maximum intercept link coverage; the number of SDN nodes required to intercept the whole network gradually increases first and then decreases gradually. This is because that when the number of N-SDN nodes is between 0 and 37 (i.e., the number of SDN nodes is between 119 and 156), though the increase of N-SDN links results in the decrease of the degree of some SDN nodes, the total number of SDN links does not decrease significantly. Thus, more SDN nodes are needed to intercept the same number of links. Accordingly, the number of SDN nodes required to intercept the whole network increases. While when the number of N-SDN is between 38 and 156 (i.e., the number of SDN is between 0 and 118), the number of SDN links greatly decreases with the increasing number of N-SDN nodes, so the number of SDN nodes needed to achieve maximum intercept link coverage also decreases gradually. Moreover, when all nodes in the network are N-SDN nodes, all links are N-SDN links, and thus the minimum vertex cover set *P* is empty (i.e., numP = 0). To sum up, according to Figure 4 and Figure 5, we only need 69~95 SDN nodes to achieve 80.53~100% intercept link coverage of the whole interception system when the number of SDN nodes in the whole hybrid SDN is between 99~156. 

Next, we will study and analyze the influence of three different approaches and three SDN interception models on intercept link coverage (ILC) as shown in Figure 8. From the figure, we can see that ILC of PA and HA with RMVCA is higher than that of RA without RMVCA in general, and ILC of PA and HA are relatively close, whether using T model, ECMP-T model or Fermat-point interception model. Additionally, compared with RA, PA and HA can significantly improve the intercept link coverage when the number of N-SDN nodes is between 0 and 60 (i.e., the number of SDN nodes is between 96 and 156). And this improvement decreases with the decrease of SDN nodes. 

Meanwhile, another conclusion we can make is that the three SDN interception models have nearly the same intercept link coverage. In other words, the intercept link coverage (ILC) has no relationship with SDN interception models and the SDN interception models have little impact on ILC.

#### 5.3.2. TTC

Using RMVCA, we will analyze the impact of RMVCA on the efficiency of deploying intercept access points in whole hybrid SDN. In many experiments, we run three SDN interception models to deploy IAPs in three approaches during which the shortest paths need to be calculated, and thus the total times of calculating the shortest path (TTC) is different. In order to evaluate the performance of RMVCA, we employ TTC as its most important performance metric. We predict that RMVCA can improve the efficiency of deploying IAPs (i.e., reduce the total deployment time). The experimental results are shown in Figure 9. 

From Figure 9, we can see that TTC of Fermat-point interception model is the highest whether in PA, RA or HA, namely, running Fermat-point model may take the longest time to deploy IAPs. In addition, TTC of T model and ECMP-T model is similar and is far lower than that of Fermat-point model. Therefore, in terms of the efficiency of deploying IAPs, T model and ECMP-T model are better than Fermat-point model.

Also, Figure 9 show the impact of three approaches in three SDN interception models on TTC under CRN topologies. From the figure, we can see that compared TTC in PA and RA, TTC in HA is the lowest, whether running T model, ECMP-T model or Fermat-point model in hybrid SDN. Namely, HA is the best approach in terms of the efficiency of deploying IAPs based on thorough analysis and comparison. 

Meanwhile, we also can see that TTC in PA is the highest and thus PA is the most undesirable approach. Considering that TTC is the most important performance metric of RMVCA, we can abandon PA. According to Figure 9, we can conclude by calculating that with the increasing number of N-SDN nodes (i.e., with the decreasing number of SDN nodes) in hybrid SDN, HA can significantly improve the deployment efficiency of intercept access points for the reason that compared with RA, HA can decrease TTC on average by 41.14%, 44.07%, 53.32% respectively in T model, ECMP-T model and Fermat-point model. In conclusion, PA is the most undesirable approach that should be abandoned. While HA is the best approach in terms of the deployment efficiency of IAPs.

#### 5.3.3. MILC, TOC and UQoS

After deploying the best intercept access point (IAP; I), the interception center (the law enforcement agencies; L) hopes to receive the data intercepted by the intercept access point with the minimum cost (i.e., the minimum cost or hop-count from the intercept access point (I) to the interception center (L); MILC). Therefore, MILC is one of the most important performance metrics of lawful interception system. In addition, the network operators are most concerned about the total cost of running intercept operation in global network (i.e., TOC) which is the prominent performance metrics of lawful interception system. Meanwhile, running different SDN interception models to deploy intercept access point may lead to the different selection of the best intercept access point (namely the placement location of IAP differs) and the different amount of calculation, thus affecting the quality of service of normal user’s data stream (UQoS). Thus, UQoS is also one of the important performance metrics of lawful interception system. In a word, MILC, TOC and UQoS are of great significance for the Law Enforcement Agencies, the network operators and the users, respectively. Focused on three hybrid SDN topologies CRN, NSFNet and COST 239, we study and analyze the impact of running three different SDN interception models to deploy the best IAPs on MILC, TOC, and UQoS of whole lawful interception system under different number of SDN nodes. The experimental results of the three topologies are shown in Figure 10a–c.

From the figures, we can see that MILC, TOC in T model are relatively close to the ones in ECMP-T model. And MILC and TOC consumed by ECMP-T model are lower than that of T model, so ECMP-T model is better than T model. More importantly, compared with MILC and TOC in T model and ECMP-T model, MILC and TOC in Fermat-point model are the lowest in all number of SDN nodes. In other words, Fermat-point model can decrease MILC and TOC compared with T model and ECMP-T model. More specifically, compared with T model and ECMP-T model, Fermat-point model can decrease MILC on average by 13.41%, 11.11% in CRN topology, 14.91%, 8.73% in COST 239, and 19.72%, 16.04% in NSFNet, and TOC on average by 1.91%, 0.99% in CRN topology, 2.82%, 0.46% in COST 239, and 2.65%, 1.03% in NSFNet. These simulation results verify that the performance of Fermat-point model outperforms T model and ECMP-T model and thus Fermat-point model is the best SDN interception model in terms of MILC and TOC.

Meanwhile, from the figures, we can see that no matter in CRN, COST 239 or NSFNet, ECMP-T model and T model have the same UQoS. In other words, T model and ECMP-T model have little impact on the transmission quality of traffic normally accepted by users and on deployment efficiency of IAP. According to the principle of T model and ECMP-T model, we know the simulation results in three hybrid SDN topologies are consistent with the theory, so these results are true and reliable. In addition, we can also clearly observe from the figures that UQoS in Fermat-point model is higher than the one in T model and ECMP-T model, which means that Fermat-point model slightly affect UQoS. Thus, Fermat-point model has poor performance in terms of UQoS.

#### 5.3.4. NFD

Due to the hybrid SDN topologies where N-SDN nodes cannot be selected as IAP, not every combination of S, D and L can successfully deploy intercept access point. We count the total number of failures to deploy IEP (i.e., NFD), to evaluate the performance of SDN interception models. The statistical results are shown in Figure 11a–c. We can clearly observe from the figures that in the three hybrid SDN topologies, the total number of failures to deploy IAPs (NFD) in Fermat-point model is the least compared with NFD in T model and ECMP-T model, which means that Fermat-point model has a high success rate to deploy intercept access point. More specifically, compared with T model and ECMP-T model, Fermat-point model decreases NFD on average by 88.21%, 86.87% in CRN topology, 76.9%, 74.68% in COST 239, and 67.53%, 66.26% in NSFNet. To sum up, the performance of Fermat-point model outperforms T model and ECMP-T model and thus Fermat-point model is the best interception model in terms of NFD.

## 6. Conclusions

In this paper, we proposed an improved equal-cost multi-path shortest path algorithm (ECMP-Dijkstra) and accordingly proposed three SDN interception models T model, ECMP-T model, and Fermat-point model, to deploy the best intercept access point reasonably in three real-world hybrid SDN topologies. Subsequently, we proposed a restrictive minimum vertex coverage algorithm (RMVCA) to intercept the whole interception system with the least SDN nodes, and to optimize the deployment efficiency of intercept access points and improve the intercept link coverage, so as to optimize the performance of the whole intercepting system. According to RMVCA, we analyze the effect of different SDN node ratios on the intercept link coverage and the minimum vertex coverage set. Considering the intercept link coverage and the minimum vertex coverage set, we found a suitable SDN node ratio for deploying intercept access points reasonably, namely, to intercept the whole hybrid SDN with the least SDN nodes.

Based RMVCA, we put forward three approaches PA, RA, and HA for experiments, and compared the three experimental approaches. The experimental results show that HA is the best approach, which can significantly optimize the efficiency of deploying intercept access points (i.e., optimize TTC) and improve the intercept link coverage of the whole hybrid SDN.

By the way, we analyzed the influence of three SDN interception models on various performance metrics of lawful interception system using three real-world topologies. The simulation results reveal that the three SDN interception models have little effect on the intercept link coverage, and T model and ECMP-T model have no effect on user’s traffic transmission quality. Compared with T model and ECMP-T model, Fermat-point model is the best interception model for the reason that Fermat-point model can make MILC, TOC, NFD the lowest by sacrificing a small part of user’s traffic transmission quality (UQoS) and deployment time (TTC), intercepting the whole hybrid SDN at dramatically lower costs. 

This paper has not considered the traffic bottleneck (link capacity) problem but has proposed the deployment and optimization strategy of intercept access points that pave the way for the future work that joint deployment of IAPs and LEAs in H-SDNs based on the consideration of the traffic bottleneck problem.

## Figures and Tables

**Figure 1 sensors-21-00428-f001:**
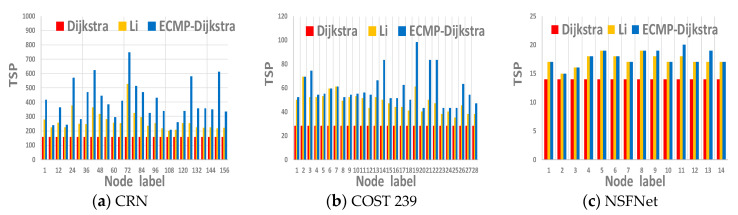
The impact of three shortest path algorithm on TSP in three topologies. TSP. The higher, the better.

**Figure 2 sensors-21-00428-f002:**
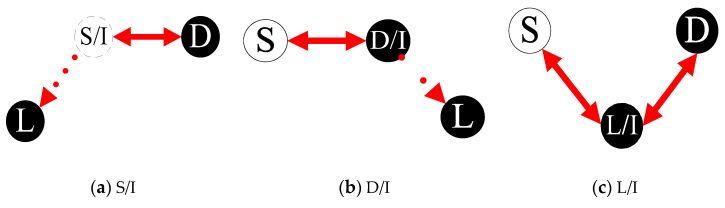
Legacy interception models.

**Figure 3 sensors-21-00428-f003:**
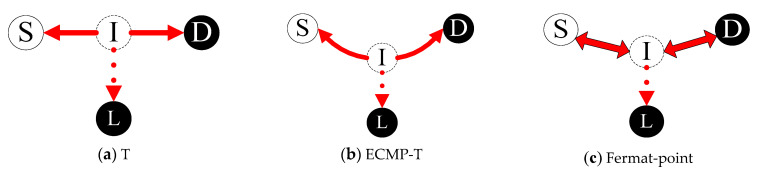
SDN interception models.

**Figure 4 sensors-21-00428-f004:**
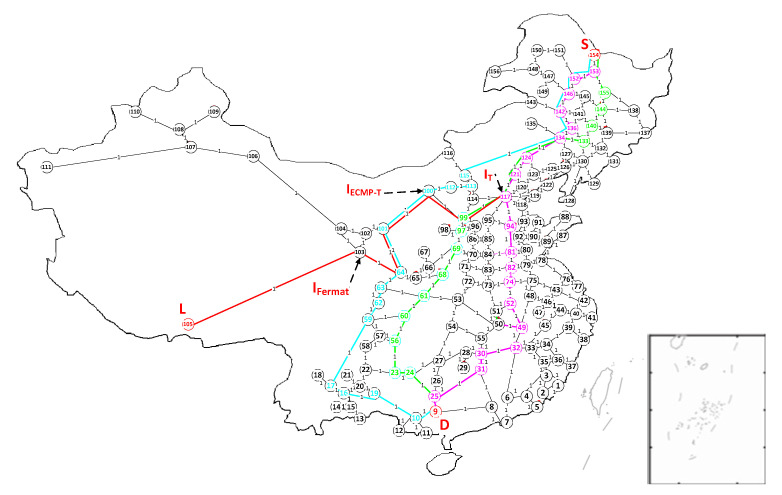
China’s 156 major railway nodes network (China Railway Network; CRN).

**Figure 5 sensors-21-00428-f005:**
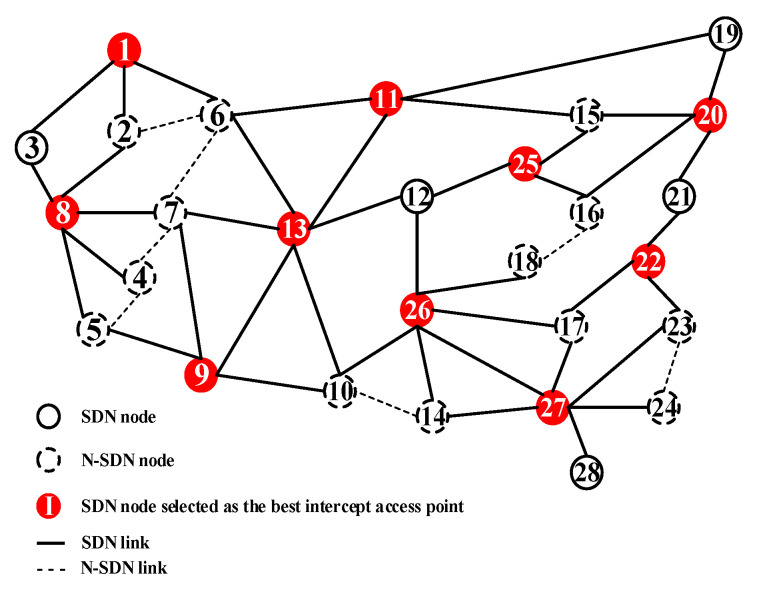
Hybrid SDN covered by minimum vertexes.

**Figure 6 sensors-21-00428-f006:**
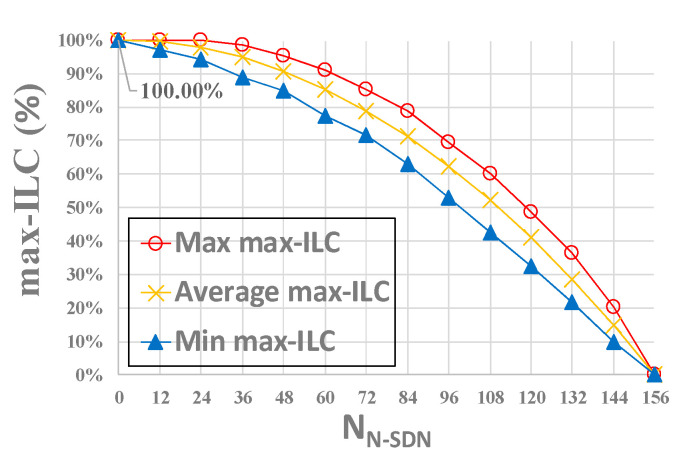
The influence of different numbers of N-SDN nodes on max-ILC.

**Figure 7 sensors-21-00428-f007:**
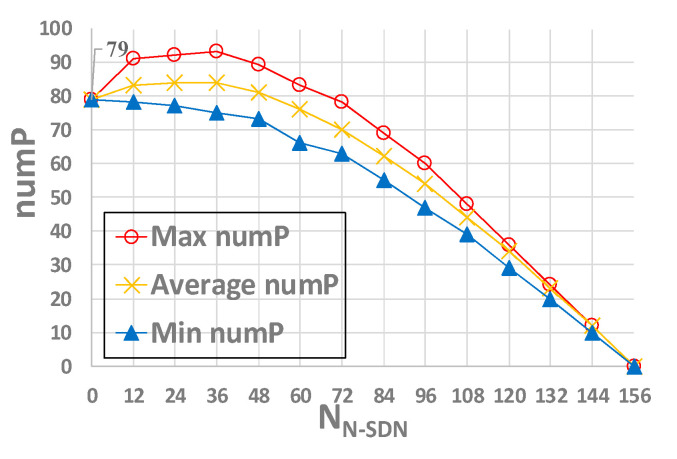
The influence of the number of N-SDN nodes on numP.

**Figure 8 sensors-21-00428-f008:**
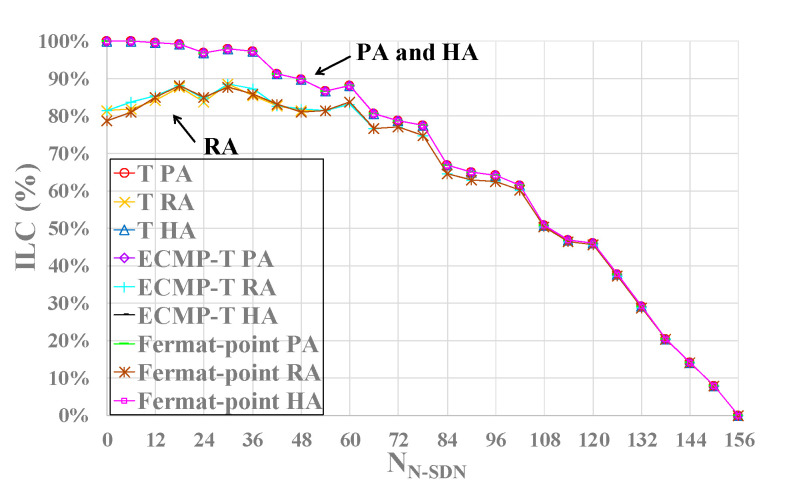
The impact of three approaches in three SDN interception models on ILC under CRN topologies. ILC. The higher, the better.

**Figure 9 sensors-21-00428-f009:**
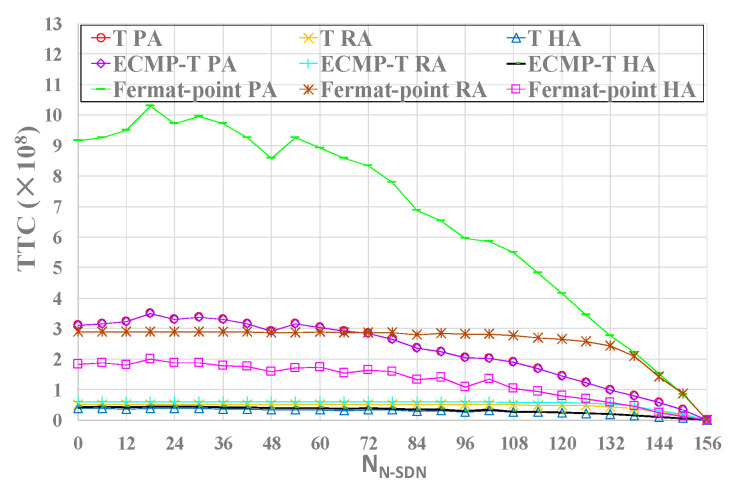
The impact of three approaches in three SDN interception models on TTC under CRN topologies. TTC. The lower, the better.

**Figure 10 sensors-21-00428-f010:**
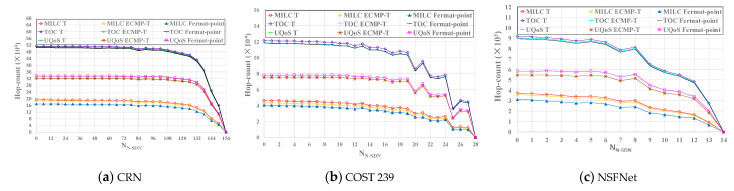
The impact of three SDN interception models on MILC, TOC, UQoS under three topologies. Hop-count. The lower, the better.

**Figure 11 sensors-21-00428-f011:**
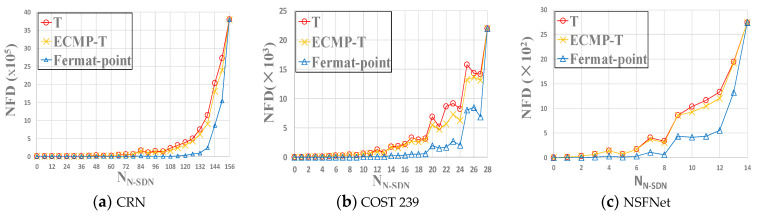
The impact of SDN interception models on NFD under three topologies. NFD. The lower, the better.

**Table 1 sensors-21-00428-t001:** Comparisons of related works.

Lawful Interception	[31]	[32]	Our Proposed
SDN based	No	Yes	Yes
Cost	Very High	Low	Low
The shortest path algorithm	[6]	[7,8,9,10,11,12,13,14]	Our proposed
Time-Space complexity	Low	Medium	Medium-Low
The number of ECMP	One	The most	All
Minimum Vertex Cover	[33]	[20,21,22,23,24,25,26,27,28,29,30]	Our proposed
Time-Space complexity	Low	Medium	Medium-Low
Results	Very bad	Near-optimal	Near-optimal

**Table 2 sensors-21-00428-t002:** Notations.

Notation	Meaning
*N_SDN_*	the SDN nodes selected randomly from all nodes in H-SDN
*S, D, L*	the source and the destination and the interception center or the LEA
*I*	the set of the best intercept access points
*sp_S,D_ or sp_S-D_*	the shortest path from node *S* to node *D*
*N_S,D_*	the set of nodes in the shortest path *sp_S,D_*
*SN*	the set of SDN nodes
*i*	the SDN node or SDN devices
*j*	the index of the j-th element of a vector
*h(i)*	the set of hop-count, *h(i)* denotes the hop-count or cost of node i
*numh(j)*	the set of costs, *numh(j)* denotes the cost of the j-th element
*inh(j)*	*in**h(j)* denotes the node with the index of j
*minhops*	the minimum cost or hop-count
*β*	the maximum index
*N*	the number of nodes
*hops(i,j) or hops_i-j_*	the minimum hop-count or cost from node i to node j

## Data Availability

The data are not publicly available due to their containing information that could compromise the privacy of research participants.

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
