# Peer review of "On the Optimal Lawful Intercept Access Points Placement Problem in Hybrid Software-Defined Networks"

_sensors, 2021, doi:10.3390/s21020428_

Round 1

Reviewer 1 Report

Authors propose Software Defined Networks technology for next generation lawful interception technology. They evaluate an improved equal-cost multi-path shortest path algorithm (ECMP-Dijkstra) in three SDN interception models (T, ECMP-T and Fermat-point). By taking into account the location relevance of all intercepted targets and the operation and maintenance cost of operators from the global perspective, they propose a restrictive minimum vertex cover algorithm in hybrid SDN (RMVCA). They evaluate their proposals in three real-world topologies. They can reasonably deploy the best intercept access point and intercept the whole hybrid SDN with the least SDN nodes, as well as significantly optimize the deployment efficiency of IAPs and improve the intercept link coverage in hybrid SDN. 

Centralized solutions are costly 

Authors are asked to add an explanation on what other considerations must be taken into account besides location, deployment of intercept access point (IAP), route selection of intercept, the minimum cost of intercept, the minimum number of intercept access points, etc. in order to implement lawful interception with full SDN. 

Author stated that centralized solutions are costly, please compare SDN-based with respect to centralized-based solutions. 

Fig 2 and 3 are very close and can be confused 

The paper is well written and explained, extensively argued and sometimes difficult to follow.  

Please delete minor typos (striketrough text)

Author Response

Dear Reviewer:

    My answer is in the attachment.

    Best regards!

Reviewer 2 Report

The manuscript presents an equal-cost multi-path shortest path algorithm for SDN environments. The results are promising. I encourage to improve plots quality and correcting the commented parts. 

Author Response

(The authors gave the same response as above.)

Reviewer 3 Report

The paper studies an improved equal-cost multi-path shortest path algorithm and proposes three SDN interception models: T interception model, ECMP-T interception model and Fermat-point interception model. The paper is well prepared and presented. However, there are a few points that the authors need to consider and fix as follows:

  1. The abstract contains a lot of abbreviations, please remove abbreviations from the abstract and leave only the full terms. You can add abbreviations when they are mentioned in the paper.
  2. Each point of the contributions in Sec 1 is very long and not focused, please rephrase them.
  3. You need to expand the related work section from the SDN point of view. You may look at these relevant and recent work: https://ieeexplore.ieee.org/abstract/document/8624382 and https://ieeexplore.ieee.org/abstract/document/8969591
  4. Also, please add a table at the bottom of the related work section to summarise the features of the current work and highlight the new features added by your proposal.
  5. How did you specify the number of SDN nodes in Fig 6-11?
  6. Please critically and technically discuss the proposed work such that you pave the way to the future work.
  7. Please proofread the paper to resolve any typographical issues.

Author Response

(The authors gave the same response as above.)

Reviewer 4 Report

Authors proposed an interesting LI-oriented approach to be applied in SDN-like large and heterogeneous networks, thus providing experimental results estimated through a simulation activity on railway connections. The paper has merit and presents an useful approach which can be adopted in real networks, so authors are encouraged to present (maybe as future work) a real deployment, with real results. Moreover, an analysis on which could be the best option for data routing (in case the intercepted traffic needs to be re-routed in such a way) to be adopted (e.g., IP, MPLS, etc.). Finally, some results need to be re-drawn, as they are in low quality.

Author Response

(The authors gave the same response as above.)
